# Measuring Residential Satisfaction in Historic Areas Using Actual–Aspiration Gap Theory: The Case of Famagusta, Northern Cyprus

Tina Davoodi [1], Balkiz Yapicoglu [1,*] and Uğur Ulaş Dağlı [2]

[1] Faculty of Design, Department of Architecture, Arkin University of Creative Arts, Kyrenia 99300, North Cyprus
[2] Faculty of Architecture, Eastern Mediterranean University, via Mersin 10, Famagusta 99450, Turkey
[*] Correspondence: balkiz.yapicioglu@arucad.edu.tr

**Abstract:** Although historical areas have significant unique architectural, historical and cultural values, and urban patterns, the physically degraded/damaged and deteriorated urban fabric of historical environments does not completely fulfill the contemporary needs of residents, which leads to low levels of resident satisfaction. As a result, this study examines the factors affecting residential satisfaction in a historic area to enhance satisfaction. Toward this objective, the present study selects the historical area of Famagusta, North Cyprus, and conducts a comprehensive survey among 129 households. Furthermore, the present study assesses residential satisfaction by incorporating socio-demographic, household environment, local historic housing renovation rules, and sense-of-place factors as well as applying the logit regression approach to find reliable results. Moreover, this study performs bootstrap Least Absolute Shrinkage and Selection Operator (LASSO) logistic regression to rank the importance of variables instead of relying on the size of estimated coefficients. To the best of our knowledge, this is among the first studies to conduct this nexus and the results could significantly contribute to the literature. Remarkably, the results reveal that the residential environment, local historical housing renovation rules, and sense of place have significant and positive effects on residential satisfaction, implying that these factors have a significant role in raising residents' satisfaction levels.

**Keywords:** residential satisfaction; historic environment; sense of place; walled city; absolute shrinkage and selection operator (LASSO)

## 1. Introduction

There has been increasing interest in residential satisfaction (RS) among scholars in recent years, with this concept being a focus of housing research and urban studies [1,2]. It is often used to evaluate residents' perceptions of whether their housing and environmental needs, expectations, and desires are being met. For example, Campbell [3] defined RS as the perception of the gap between residential expectation and reality, while Morris and Winter [4] argued that it depends on the compatibility between real housing satisfaction and housing norms. Galster [5] suggested that residential conditions among households are influenced by individuals' needs and desires, and Amérigo and Aragonés [6] emphasized the role of physical and social characteristics in measuring RS in housing and neighborhood studies. On the other hand, Hunt [7] argued that satisfaction is not an emotion, but rather an assessment of emotion. Ibem and Amole [8] also reported that RS is often used to evaluate residents' perceptions of whether their needs, expectations, and aspirations are being met by their housing units and the surrounding environment.

Consequently, a great deal of research has been conducted on residential satisfaction (RS) in the past few years, with many scholars focusing on three main theories: "housing needs," "housing deficit," and "psychological construct" [1,9–13]. These theories have

been explored both theoretically and empirically. Empirical studies have found that factors, such as socio-economic characteristics, housing features, neighborhood environment, social environment, and housing support services, are significant predictors of RS [9,14–16]. In addition, more recent research has shown that the sense of place and the actions of local authorities, such as housing policymakers and urban planners, can also impact RS [17–19]. There is a considerable body of literature on the determinants of RS in various countries and contexts, such as private housing [20], neighborhood [21–23], public housing [19,24–26], and mass housing [27], but limited attention has been paid to investigating RS in historic environments, particularly in emerging countries. This study aims to fill this gap by examining the factors that influence RS in a historic area.

The historic environment is characterized by the intersection of intangible elements, such as memories, with tangible elements, such as buildings, and represents the tangible evidence of past human activity and cultural heritage. However, the physical degradation or deterioration of the urban fabric in these environments can have a negative impact on residential satisfaction (RS). This is particularly true in the case of historical residential areas, which may have unique and distinctive cultural characteristics that could otherwise contribute to higher levels of satisfaction among residents. Old buildings in historic environments may also have poor structural conditions and outdated living environments, which can further decrease RS. Vehbi and Hoskara [28] argued that historical areas have suffered from significant decay in physical, social, and economic terms due to declining economic activity, leading to lower levels of RS, reduced ownership and rental rates, and, ultimately, mobility of residents. It is also important to note that the aspired residential environment of residents in historic areas may differ from that of other groups, and the effects of various factors on RS may be different for this particular group of residents living in historical environments.

This study aims to investigate the factors that impact residential satisfaction (RS) in the historic environment of the Walled City of Famagusta in North Cyprus, an area that has received little attention from researchers to date. The unique characteristics of this historic environment and the lack of empirical research in this area are the primary motivations for conducting this study. To the best of the authors' knowledge, this study is among the first to specifically examine the predictors of RS in the Walled City of Famagusta. While Davoodi and Dağlı [29] identified important factors related to the sustainability of the Walled City through a "Marginal improvement priority" approach, they did not measure the gap between actual reality and aspirations through the level of satisfaction of residents, which this study intends to do. The Walled City of Famagusta has distinctive characteristics, including its organic urban pattern with narrow streets and low-rise dwellings that were developed over various historical periods and still show evidence of its history [30]. In the past two decades, the municipality and the United Nations Development Project (UNDP) have worked to revitalize the Walled City in an effort to protect its cultural heritage, prevent deterioration, and sustain its livability and economy.

To create an empirical satisfaction model for the historical area of the Walled City, this study applies the actual–aspiration gap theory, as proposed by Galster [5], and follows prior empirical studies using "the residential environment," "local historical housing renovation rules," and "sense of place" (SOP) as the four pillars of residential satisfaction (RS). Additionally, "the neighborhood facilities," "the environmental features," "the social environment," "the housing conditions," "the housing features," and "the housing support services" are used as sub-pillars to measure the "residential environment" pillar. Similarly, "the place," "the place attachment," and "the place dependence" sub-pillars are employed to measure the SOP pillar. The selection of these pillars is based on the specific and unique characteristics of historical areas, which distinguish historical environments from other residential areas, with the following objectives in mind:

- How do the residential environment, local historical housing renovation rules, and SOP pillars impact RS?
- How do the sub-pillars of the residential environment and SOP pillars impact RS?

- How do the components of the residential environment, local historical housing renovation rules, and SOP pillars impact RS?

The present study aims to shed light on how the various aspects of the pillars impact RS in the historical area of the Walled City through the use of an empirical satisfaction model.

The authors believe that this study can contribute to the existing research on the topic in several ways. First, the present study examines the factors of residential satisfaction (RS) in a historic environment, which is an area that has been understudied in the literature. Second, this study comprehensively examines the predictors of RS, including those that have not been used in recent RS estimation models, such as socio-demographic, residential environment, local historical housing renovation rules, and sense of place (SOP). Third, this study uses the appropriate methodology of logit models because the estimated results via multiple linear regression (OLS technique) in the presence of a binary dependent variable are not robust [31]. Lastly, the distinctive contribution of this study is that it employs a bootstrap LASSO logistic regression to rank the importance of determinants, rather than using the magnitude of coefficients [32,33]. The findings of this study also provide some insights into the RS literature for further studies in other historical areas and the outcomes may also be useful for housing policy decision makers, urban planners, and municipalities as they work to better understand the needs and expectations of residents in historic environments and help to reconcile any discrepancies between these needs and expectations. Furthermore, it can be considered a crucial step in the process of sustainable development. Reinforcing the positive elements and eliminating the negative elements of RS play significant roles in enhancing the sustainability of the Walled City.

The remainder of the article is organized as follows: In Section 2, a review of the related literature is provided. The conceptualization of the study is presented in Section 3 with hypotheses. Section 4 describes the models and methodology for the study, along with a description of the study area, and the results are presented in Section 5, followed by the discussion and conclusion in Sections 6 and 7. Additionally, Figure 1 illustrates the research process of this study.

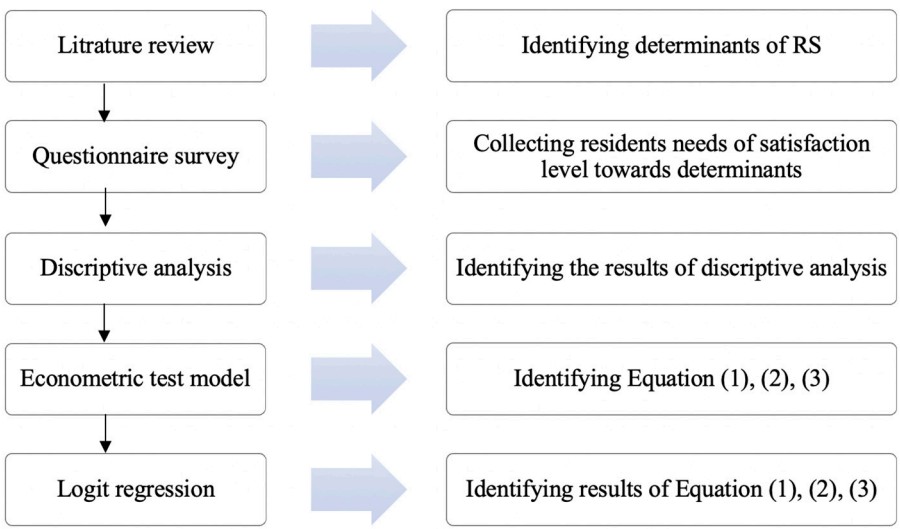

**Figure 1.** Research process.

## 2. Literature Review

Residential satisfaction (RS) has been widely debated by scholars and researchers in the fields of sociology, planning, urban studies, geography, and psychology [34–38]. In a seminal study, Galster [5] highlighted that most studies of RS are based on two opposing empirical approaches: the "purposive" approach and the "actual-aspiration gap" approach. The purposive approach views RS as a measure of the extent to which a residential environment helps residents achieve their goals [5,34,39,40]. In this approach,

researchers have mainly focused on goals or activities related to the attributes of the physical environment. However, in the actual–aspiration gap approach, RS is viewed as an indicator of the gap between what residents want and aspire to have and what they currently have in terms of their housing conditions [5]. Some authors, such as Ibem and Abole [8] and Chaudhury and Rowles [41], also noted that RS is a measure of residents' perception of the adequacy of their residential environment in meeting their needs, expectations, and aspirations. In the literature, studies have used both objective and subjective measurements to assess RS [35,42]. Furthermore, Amole [43] pointed out that objective attributes of the residential environment, when evaluated by residents, become subjective. Objective measurement of RS involves the actual physical characteristics, facilities, services, and environment, while subjective measurement is linked to the psychological aspect of humans and measures perception, emotions, attitudes, and aspirations [5,44].

Multiple studies have explored the relationship between RS and residential mobility. Speare [45] argued that satisfaction plays a mediating role between individual and residential variables and residential mobility. In the literature, studies suggest that RS with regard to the house and neighborhood can impact mobility behaviors and a higher actual–aspiration gap between residents' needs and expectations may lead to relocating and mobility of residents. For example, Diaz-Serrano and Stoyanova [46] found that RS has a strong mediating effect on mobility propensity. Other studies [47–49] found that RS is an important factor that affects residents' intention to move. Findings of many studies highlighted that dissatisfaction with social problems impact the mobility of residents, as well as residential satisfaction [50]. Similarly, Jia and Lei [51] found that RS affects the intentions to move among both locals and migrants. However, Zhang and Lu [52] found that lower RS does not necessarily lead to high residential mobility.

Additionally, a portion of the literature has examined the significant determinants of RS empirically. For instance, Savasdisara et al. [53] found that residents are generally more satisfied with their dwelling units than with environmental facilities. Fang [54] found that unit size has a positive effect on RS. In contrast, Kaitilla [55] and Nurizan [56] found that residents are dissatisfied with the size of houses. On the other hand, Husna and Nurijan [57] found that residents are dissatisfied with dwelling unit characteristics, while Ha [58] found the opposite. Oh [59] and Jiboye [60] found that residents were dissatisfied with public facilities and management policies, respectively. In recent studies, researchers have found the impact of different determinants of RS in different countries. For example, Addo [61] found that dwelling characteristics have a negative influence on the RS of respondents in Ghana. Cao and Wang [62] suggested that improving parks and open spaces, neighborhood safety, and neighborhood appearance is important for enhancing RS of existing residents in Twin Cities. Gan and Zuo [63] found that factors in the domains of dwelling features, dwelling facilities, and housing policies had a positive contribution to residential satisfaction, while factors in the domains of public facilities and neighborhood environment had a negative contribution in Chongqing.

Furthermore, Appendix A, Table A1, presents a summary of empirical studies on residential satisfaction (RS). As shown in Table A1 and explained here, RS has been extensively studied in the context of public housing [19,25,64], urban housing [17], planned communities [18], private low-cost housing [20], and double-story terraces [9]. These studies have generally examined RS in terms of various dimensions, such as socio-demographic factors and residential environment (including neighborhood facilities, housing features, environmental features, rules, and sense of place), and have been conducted in a variety of countries, including Nigeria [40,64], Malaysia [20,44], Turkey [17,65,66], Italy [67], Australia [18], and Iran [19,25].

According to the studies summarized, housing characteristics and neighborhood environments have consistently been found to be significant factors in determining residential satisfaction (RS) [8,15,68,69]. Additionally, research has shown that neighborhood facilities [9,20,44], socio-economic and demographic profiles [69], local authorities [17], social environment [17,44], accessibility [65], residence comparison and public housing allocation

schemes [15], and housing services [20,40] all impact RS. Some studies have also found that sense of place can affect RS [18,19].

Several studies have focused specifically on examining the determinants of residential satisfaction (RS) in old buildings and historical areas. Phillips et al. [70] found that ventilation and sunshine are important factors impacting RS for older dwellings in Hong Kong. Furthermore, Erdogan et al. [17] found that social and environmental living conditions have a positive effect on overall housing satisfaction in historic neighborhoods in Edirne, Turkey. Temelová and Dvořáková, [71] showed that elderly residents are moderately satisfied with their residential environment in the historic core of Prague city center. Zhang and Lu [52] found that, despite lower dwelling conditions, residents in traditional neighborhoods were generally more satisfied with their physical environment compared to residents in redeveloped neighborhoods. Li and Song [72] found opposite results.

While there is a large body of literature on RS, there are relatively few studies that have specifically focused on examining RS in historical areas. This study aims to contribute to this body of research by investigating the determinants of RS in the historical area of the Walled City of Famagusta, North Cyprus. In developing the conceptual framework for this study, the following dimensions were identified.

## 3. Conceptualization

### 3.1. Determinants of RS

After a thorough review, this study chooses and presents the factors of RS that are used in this study, and these are listed in Table 1. In line with the previous studies listed in Table A1, this study selected the pillars of RS as socio-demographic characteristics, residential environment, and SOP. Given the specific characteristics of the historic environment, this study also included the local historical housing renovation rules pillar. As shown in Table 1, the residential environment pillar is divided into the sub-pillars of neighborhood facilities, environmental features, social environment, housing conditions, housing features, and housing support services. Additionally, SOP is broken down into the sub-pillars of place identity (hereafter referred to as PI), place attachment (hereafter referred to as PA), and place dependence (hereafter referred to as PD). Table 1 also lists the specific components of each pillar and sub-pillar that are used in this study.

#### 3.1.1. Socio-Demographic Characteristics

Demographic and socio-economic characteristics have been shown to significantly impact RS [73]. In a seminal study, Adriaanse [69] found that characteristics of households, such as age, marital status, gender, race, and household size, influence the level of RS. Similarly, Newman et. al [74] and Lu [75] found that factors, such as age, home ownership, income, and duration of residence, impacted RS. Numerous studies have also shown that factors, such as gender, income [20,76], age [8,77], marital status and education level [20,78], and length of stay [79], impact RS. According to the literature review, socio-demographic determinants, such as age, reflects the way people view their roles in society; gender significantly affects an individual's way of thinking; marital status creates many shared opportunities and obligations; individuals with different income levels may display different residential satisfaction on similar accommodation environments; higher income might provide greater security, more freedom, and financial advantage; length of stay affects resident preferences to stay in the community and, therefore, their responses to the surveys. In this study, the effect of socio-demographic characteristics is measured using variables, such as gender, age, monthly income, home ownership, length of residency, ethnicity, household size, marital status, education, employment sector, and number of bedrooms. Based on the reviewed literature, it is expected that socio-demographic characteristics will impact RS.

**H1:** *There is a positive or negative significant relationship between socio-demographic characteristics and RS.*

**Table 1.** Residential satisfaction determinants.

| Pillar | Sub-Pillar | Components | Source |
|---|---|---|---|
| Socio-demographic characteristics | ____ | Gender; Age; Monthly income; Home ownership; Length of residency; Ethnicity; Household size; Marital status; Education, Employment sector; Number of bedrooms | [8,9,24,25,38,44,64] |
| Residential environment | Neighborhood facilities: (Accessibility to function area and public facilities) | Accessibility to centrality (shopping center, city center (Namik Kemal Square, Salamis Road), Job, market); Accessibility to education (kindergarten, primary schools, high schools); Accessibility to open areas (public and private parking areas, walking areas, sport center, green space and views, sea); Accessibility to health center; Accessibility to public transport (bus, taxi); Accessibility to public facilities (Park, place of worship, police station, firefighting station, bank, recreational areas, closed sport center) | [9,15,20,44,63,65] |
| | Environmental features | Maintenance of the environment (open areas, green areas); Night lighting; Traffic density; Cleanliness | [1,65] |
| | Social environment | Neighborhood relations; Community cohesion/relations; Level of crime; Level of security; Noise | [15,44] |
| | Housing conditions | Water pressure; Quality of doors; Quality of floors; Quality of walls; Quality of windows; Quality of interior painting; Quality of exterior painting; Lighting of stairwell | [64] |
| | Housing features | Design/layout of the interior space; Design of bath and toilet; Location and size (living room, kitchen, bathroom, toilet); Number (bedroom, bathroom, toilet); Natural lighting and ventilation; Materials; Privacy; Thermal insulation | [20,40,44,64,69] |
| | Housing support services | Pipe and electrical repair services; Garbage collection; Internet; Fire system; Street lighting | [2,9,40,44] |
| Local historical housing renovation rules | --- | Height of building; Percentage of land use for construction; Proportion of openings; Types of the facade materials; Renovation approval process | [17,42,44] |
| Sense of place | Place identity | Identity 1; Identity 2; Identity 3; Identity 4 | [19] |
| | Place attachment | Attachment 1; Attachment 2; Attachment 3; Attachment 4 | |
| | Place dependence | Dependence 1; Dependence 2; Dependence 3; Dependence 4 | |

Note: This table presents the determinants of residential satisfaction used in this study.

### 3.1.2. Residential Environment

RS can be explained through factors such as the physical and social environment surrounding the home, including neighbors, the neighborhood, the quality of housing, and green areas [5]. RS is a dynamic process of interaction between residents and the social and physical factors of the environment [5]. Many studies have attempted to measure the degree of satisfaction with the residential environment among residents [20,44]. Canter and Rees [34] found that the three components of the residential environment—the house, neighbors, and neighborhood—explain RS. Other studies have also argued that the concept of home extends beyond the walls of the house and encompasses elements, such as noise and traffic density [80], the physical appearance of the neighborhood [81], relations with neighbors [82], and green areas or parks [83], when considering RS.

There is a significant body of research that has examined the relationship between residential environment and residential satisfaction. For example, studies have found that poor housing conditions [64] and a lack of neighborhood facilities [20] can lead to dissatisfaction among residents. On the other hand, a well-maintained residential environment [65] and good housing features [44] can contribute to satisfaction. Additionally, factors such as the social environment [84] can impact residents' satisfaction with their living situation. Based on this literature, this study investigates the impact of several sub-factors of the residential environment on residential satisfaction, including neighborhood facilities, envi-

ronmental features, social environment, housing conditions, housing features, and housing support services.

- Neighborhood facilities [9];
- Environmental features of housing [65];
- Social environment [44];
- Housing conditions [64];
- Housing features [40];
- Housing support services [2].

This study also measures the impact of each sub-pillar of the residential environment on RS by selecting several components, as shown in Table 1. According to the literature review, it is expected that the residential environment will have an impact on RS.

**H2:** *There is a positive or negative significant relationship between residential environment and RS.*

### 3.1.3. Local Historical Housing Renovation Rules

A number of studies suggested that residential satisfaction (RS) can be understood from cognitive, affective, and conative perspectives [42]. In particular, Francescato [42] noted that management policies, rules, and practices, in addition to the physical environment and housing complex, can impact satisfaction. Erdogan et al. [17] also found that local authorities, such as housing policy makers and urban planners, can significantly impact housing satisfaction through the implementation of effective regulations and strategies. Mohit et al. [44] similarly argued that local authorities have a significant role to play in improving RS through the determination and implementation of suitable management policies. Based on the literature, this study expects that local historical housing renovation rules will have an impact on RS. In a historic urban quarter of the Walled City, specific local historical housing renovation rules have been implemented in order to preserve the characteristics of historical houses and environments. These rules may restrict the flexibility of local residents in terms of housing renovations, leading to either satisfaction or dissatisfaction. To examine the effect of local historical housing renovation rules on RS, this study collected information on existing housing renovation rules of the Walled City from the department of Antiquities law in the Turkish Republic of North Cyprus (TRNC). As shown in Table 2, local historical housing renovation rules include components, such as the height of houses, percentage of land used for construction, proportions (height and width) of openings, types of façade materials, and the renovation approval process.

**H3:** *There is a positive or negative significant relationship between local historical housing and RS within the walled city.*

**Table 2.** Respondents' demographic and socio-economic characteristics from Field Survey in 2022.

| Socio-Demographic Characteristics | Variable | Frequency (n = 129) | % | Overall Satisfied | | Overall Dissatisfied | |
|---|---|---|---|---|---|---|---|
| | | | | Frequency (n = 129) | % | Frequency (n = 129) | % |
| Gender | Male | 67 | 51.94 | 38 | 45.78 | 29 | 63.04 |
| | Female | 62 | 48.06 | 45 | 54.22 | 17 | 36.96 |
| Age | <18 | 4 | 3.10 | 3 | 3.61 | 1 | 2.17 |
| | 19–30 | 36 | 27.91 | 22 | 26.51 | 14 | 30.43 |
| | 31–61 | 69 | 53.49 | 46 | 55.42 | 23 | 50.00 |
| | >61 | 20 | 15.50 | 12 | 14.46 | 8 | 17.39 |

**Table 2.** *Cont.*

| Socio-Demographic Characteristics | Variable | Frequency (n = 129) | % | Overall Satisfied | | Overall Dissatisfied | |
|---|---|---|---|---|---|---|---|
| | | | | Frequency (n = 129) | % | Frequency (n = 129) | % |
| Monthly income (Turkish lira Ł) | <2000 Ł | 25 | 19.38 | 16 | 19.28 | 9 | 19.57 |
| | 2000–3000 Ł | 53 | 41.09 | 36 | 43.37 | 17 | 36.96 |
| | 3000–4000 Ł | 30 | 23.26 | 14 | 16.87 | 16 | 34.78 |
| | 4000–6000 Ł | 13 | 10.08 | 9 | 10.84 | 4 | 8.70 |
| | >6000 Ł | 8 | 6.19 | 8 | 9.64 | — | — |
| Home ownership | Owner | 63 | 48.84 | 48 | 57.83 | 15 | 32.61 |
| | Rental | 66 | 51.16 | 35 | 42.17 | 31 | 67.39 |
| Length of residency | <5 Years | 32 | 24.81 | 20 | 24.10 | 12 | 26.09 |
| | 5–10 | 18 | 13.95 | 9 | 10.84 | 9 | 19.57 |
| | 10–20 | 28 | 21.71 | 20 | 24.10 | 8 | 17.39 |
| | 20–40 | 25 | 19.38 | 15 | 18.07 | 10 | 21.74 |
| | >40 | 26 | 20.16 | 19 | 22.89 | 7 | 15.22 |
| Ethnicity | Turkish-Cypriot | 84 | 65.12 | 57 | 68.67 | 27 | 58.70 |
| | Non-Turkish-Cypriot | 45 | 34.88 | 26 | 31.33 | 19 | 41.3 |
| Household size (people per house) | 1 | 5 | 3.88 | 3 | 3.61 | 2 | 4.35 |
| | 2 | 32 | 24.81 | 23 | 27.71 | 9 | 19.57 |
| | 3 | 30 | 23.26 | 20 | 24.10 | 10 | 21.74 |
| | 4 | 38 | 29.46 | 25 | 30.12 | 13 | 28.26 |
| | >4 | 24 | 18.59 | 12 | 14.46 | 12 | 26.09 |
| Marital status | Single | 49 | 37.99 | 30 | 36.15 | 19 | 41.30 |
| | Married | 80 | 62.01 | 53 | 63.85 | 27 | 58.70 |
| Education | Primary school | 23 | 17.83 | 14 | 16.87 | 9 | 19.57 |
| | Middle school | 20 | 15.50 | 13 | 15.66 | 7 | 15.22 |
| | High school | 43 | 33.33 | 26 | 31.33 | 17 | 36.96 |
| | University degree | 43 | 33.34 | 30 | 36.14 | 13 | 28.26 |
| Employment sector | Private | 16 | 12.40 | 12 | 14.46 | 4 | 8.70 |
| | Public | 113 | 87.60 | 71 | 85.54 | 42 | 91.30 |
| Number of bedrooms | 1 | 15 | 11.63 | 12 | 14.46 | 3 | 6.52 |
| | 2 | 41 | 31.78 | 24 | 28.92 | 17 | 36.96 |
| | 3 | 54 | 41.86 | 32 | 38.55 | 22 | 47.83 |
| | 4 | 17 | 13.18 | 13 | 15.66 | 4 | 8.70 |
| | >4 | 2 | 1.55 | 2 | 2.41 | — | |

### 3.1.4. Sense of Place (SOP)
Definition of SOP

Norberg-Schulz [85] introduced us to a phenomenological understanding (how people receive sensory materials about the physical world and consciously process this material to find meaning) of place and the Roman Concept of Genuis-Loci, or the guardian spirit of the place. According to Norberg-Schulz [85], the concept of place is examined through our ability to connect with the physical character of a geographic setting, thereby bestowing the place with identity and meaning and states that "the spaces where life occurs are places". Many researchers after Norberg-Schulz also explored the concept of place and proposed various definitions, which do not change much in their meanings. Canter [86] defined place as an environmental experience, and later added that people's awareness of these experiences influences their behavior. Other researchers, such as Jorgensen and Stedman [87], Shamai and Ilatoy [88], and Tonts and Atherley [89], argued that people's feelings toward a place or the people in a particular place are the main factors in sense of place (SOP). In their research, Jorgensen and Stedman [87] identified three constructs of SOP: place identity (PI), place attachment (PA), and place dependence (PD). PI refers

to the cognitive relationship that individuals have with the physical environment and individuals experience in their daily interaction with the urban area surrounding them [90]. As mentioned in Zurchi and Nafa [85], cultural heritage tangible elements of PI can be identified as the physical form and functionality of the place, whereas intangible elements refer to the socio-cultural factors (i.e., activities that occur in the historical places with regards to the traditions), which this study emphasizes. Furthermore, as Proshansky stated [91], PI develops through a pattern of tendencies, including beliefs, preferences, ideals, emotions, personal objectives, and behaviours. PA, on the other hand, refers to the affective association that people have with their environment, which goes beyond cognition or judgment and also emotionally connects individuals to the past [92–94]. PD, however, is a conative link between individuals and specific spatial settings, reflecting how well a place serves an individual's needs compared to other possible places. In summary, place identity (PI), place attachment (PA), and place dependence (PD) are important components of sense of place (SOP). PI is a cognitive relationship that individuals have with their physical environment, characterized by beliefs, preferences, ideals, emotions, and behaviors. PA is an affective association with the environment, encompassing emotional responses and behavioral commitments to a place. Many researchers have argued that PI, PD, and PA overlap and may be seen as sub-dimensions of PA. If the reaction is positive, it can lead to attachment to that place. Similarly, Kyle et al., [95] identified both physical and social environments as causes of emotional responses to a specific place. Lastly, PD is determined by the link between humans and specific spatial settings [96]. The PD construct is influenced by people's needs or personal objectives. Jorgensen and Stedman [87] and Williams et al. [93] stated that PD is an indication of how well a specific spatial setting meets individuals' needs compared to other accessible and potential places.

In summary, place identity (PI), place attachment (PA), and place dependence (PD) are important components of sense of place (SOP). PI is a cognitive relationship that individuals have with their physical environment, characterized by beliefs, preferences, ideals, emotions, and behaviors. PA is an affective association with the environment, encompassing emotional responses and behavioral commitments to a place. PD is a conative link (i.e., behavioral intentions) between individuals and specific spatial settings, reflecting how well a place serves an individual's needs compared to other possible places.

Linkage between SOP and RS

Based on the literature review, this study expects that SOP (consisting of PI, PA, and PD) has a significant impact on RS. In order to measure the effect of SOP on RS, this study will use the components of SOP, as shown in Table 2. These components include place identity, place attachment, and place dependence, which are sub-dimensions of SOP. The specific components of each sub-dimension will be used to measure the effect of SOP on RS in the historic urban quarter of the Walled City.

In previous research [19,87,97], it has been shown that using a multi-dimensional approach to measure sense of place (SOP) is effective. This approach allows researchers to better understand the complexity of SOP and its cognitive, conative, and affective aspects. Studies have also shown that a multi-dimensional framework leads to more reliable results and is the best fit for the data [87]. In this approach, each component of SOP is treated as a separate construct. This approach is useful for capturing the cognitive (PI), affective (PA), and conative (PD) dimensions of people's relationships with their environment [87]. Based on the research of [87], a 12-item scale was selected to examine the impact of PI, PA, and PD on residential satisfaction (RS).

This study hypothesizes that SOP impacts residential satisfaction (RS). This hypothesis is supported by the historical context of the Walled City, which is being preserved as cultural heritage through the UNDP-PFF (see Section 4). Previous research [98–102] has indicated that the protection of cultural heritage can affect residents' SOP. For instance, Smith [98] argued that heritage both reflects the past and shapes current experiences, influencing how people perceive their environment. This, in return, can result in a sense of identity and

belonging for certain individuals or groups. Similarly, Davis et al. [100] found that some communities establish their identity through cultural heritage in order to enhance people's associated SOP. When examining the relationship between historic places and locals' SOP, Davis et al. [99] discovered that adult residents had stronger SOP when living in areas with more historic environments. These environments were found to have significant and positive relationships with locals' SOP [99]. Schofield and Szymanski [102] also demonstrated this relationship between cultural heritage and SOP. Additionally, Hawke's research [101] indicated that self-esteem, distinctiveness, and continuity are reinforced by cultural heritage, and these elements together inspire the formation of place identity (PI) [103].

**H4:** *There is a correlation between SOP and RS.*

### 3.2. A Conceptual Framework of RS

Residential satisfaction (RS) is often measured by the gap between residents' actual experiences and their aspirations or expectations. Previous research has demonstrated the impact of socio-demographic characteristics and residential environment on RS [2,40,44]. However, the impact of local historical rules and sense of place on RS has received relatively little attention in the literature. Only a few studies have examined the effect of local authorities [17] and sense of place [17,19] on RS. It is important to consider these factors in the context of historical environments, as specific renovation rules can help to preserve the characteristics of historical houses/buildings and environments [29]. Therefore, the conceptual framework for this study, shown in Figure 2, includes local historical renovation rules and sense of place, in addition to socio-demographic characteristics and residential environment.

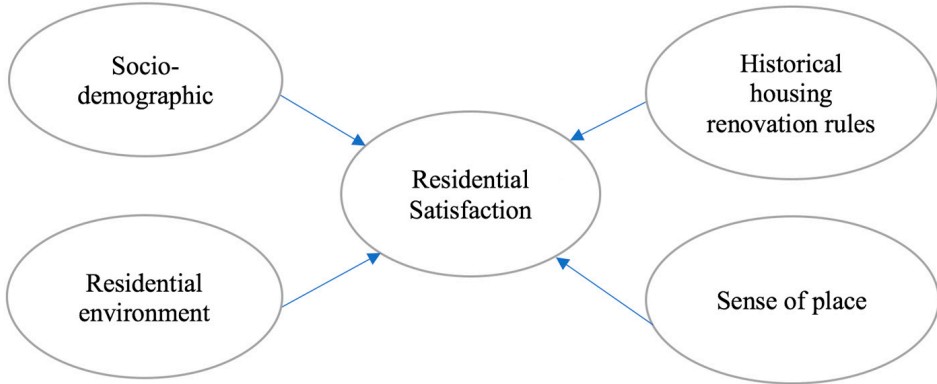

**Figure 2.** Conceptual framework of this study.

## 4. Model and Methodology

### 4.1. Study Area

Cyprus is the third largest island in the Mediterranean Sea, located 60 km south of Turkey, 322 km from Greece, and 96 km west of the Syrian coast [104]. The Walled City of Famagusta is a historic residential district situated on the eastern coast of the island. It is encircled by fortifications, consisting of Venetian-era masonry walls that have been well preserved over the years. This area is an important cultural and historical site, with a rich history dating back to medieval times (Figure 3). The Walled City of Famagusta developed throughout various historical periods (Lusignan, Venetian, Ottoman) and, during each period, it was characterized by significant development. These diverse historical backgrounds enabled the Walled City to acquire some distinctive characteristics in housing form and urban pattern. Figure 4 illustrates the locations and periods of construction of some important historic monuments (cultural heritage) of the Walled City in specific detail. Over the last two decades, there have been efforts to protect these historical houses and preserve the cultural heritage of the Walled City. For instance, local historical housing renovation rules were established by the department of Antiquities

law in the Turkish Republic of North Cyprus (TRNC). These local rules aim to renovate historical houses while preserving their traditional characteristics. Figure 5 shows samples of some historical houses by comparing images before and after renovation. The rules consist of height limitations for buildings (Figure 5a), limits on the application of façade material (Figure 5b), and proportions of openings (Figure 5c).

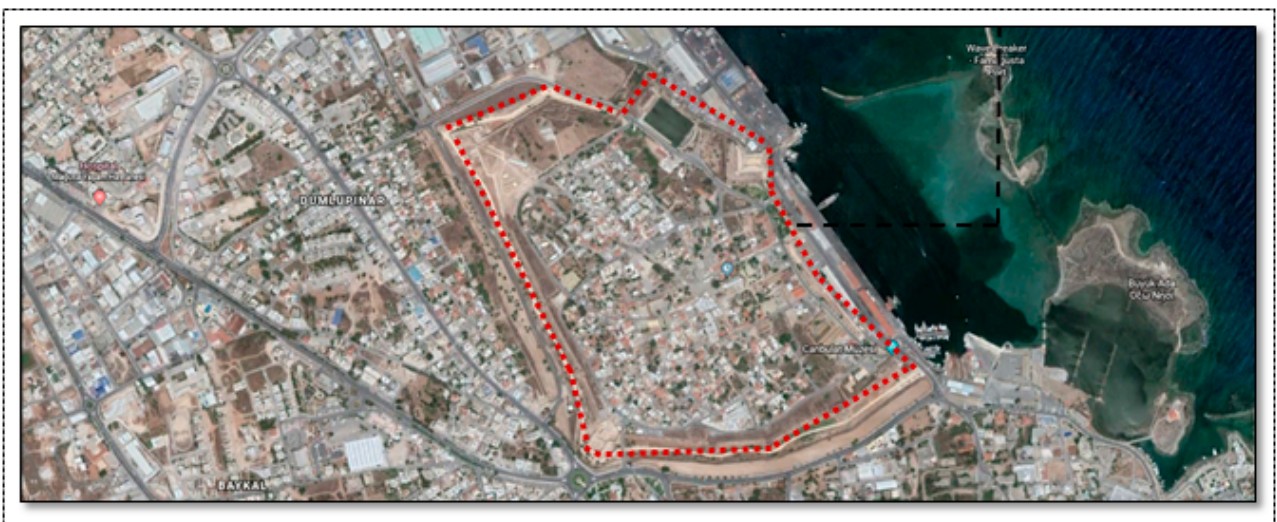

**Figure 3.** Location of Walled City; source: extracted and modified from Google Earth (2022).

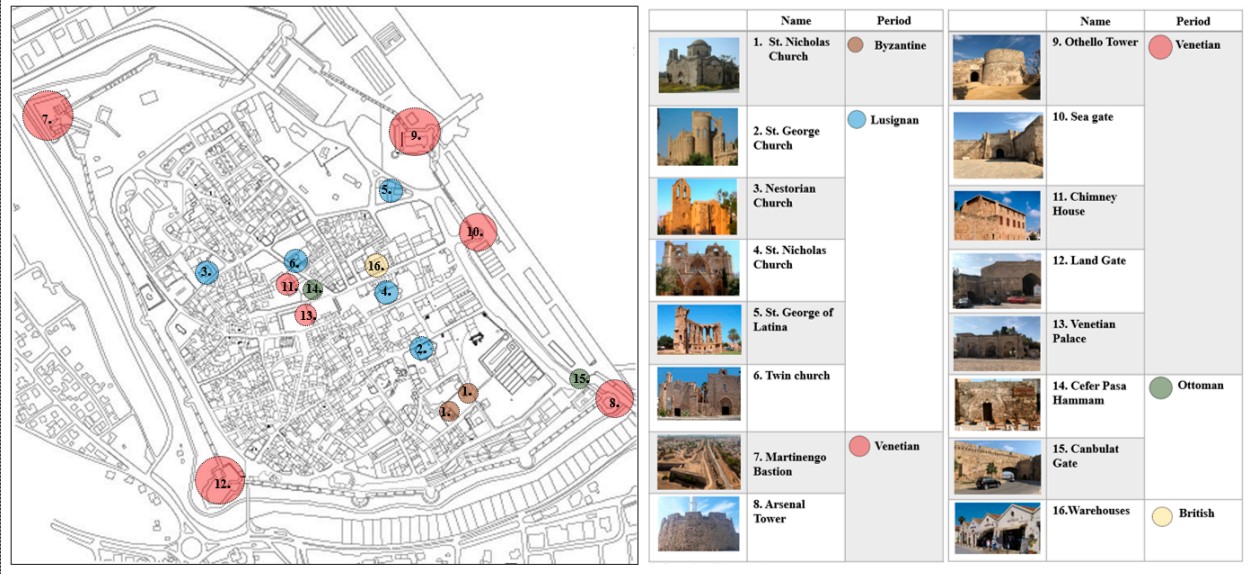

**Figure 4.** Locations and the periods of historical monuments; source: authors (2022).

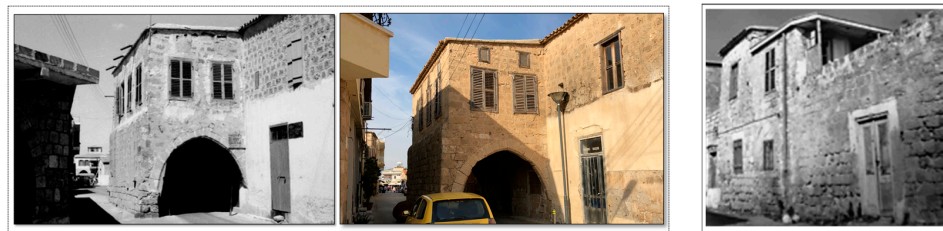
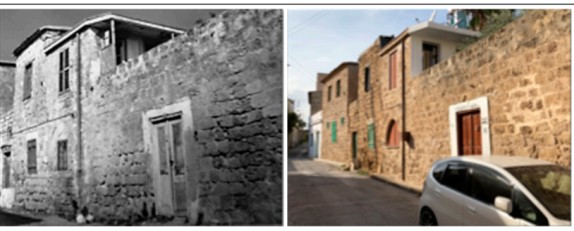

(**a**) Height of building　　　　　　　　　　　(**b**) Façade material

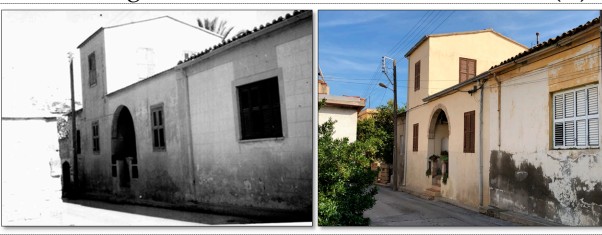

(**c**) Proportion of openings

**Figure 5.** Comparing historical houses before and after renovation. Source: photo gallery of Famagusta Walled City [105] before renovation, authors (2022) after renovation.

## 4.2. Data Collection and Methodology

To select the pilot participants, this study uses the convenience sampling technique. Administering a survey in convenience sampling is aimed at willing, geographical proximity and accessible participants, and it is best used for testing as part of hypothesis generation, gaining a sense of opinions or even as an initial pilot before further research [106]. This technique is suitable where adequate information on population size is lacking. Therefore, findings drawn may not be generalizable; however, using more respondents, the findings can be representative [106]. The data for this study were collected through the distribution of questionnaires to the residents of the Walled City. A sample of 129 households, living in different historical buildings, was determined based on the total population of the Walled City and a 95% confidence level. The total number of habitable historical houses is estimated to be 200 [107]. Participants were randomly selected through either personal visits to explain the aim of the questionnaire or by scheduling individual appointments for those who were unavailable. The survey was conducted in the evenings during the summer months of August and September 2018. The questionnaire was initially written in Turkish as it is the primary language in Northern Cyprus, but an English version was also provided for foreign non-Turkish participants. Only closed-ended questions were used in the survey. The questionnaire was divided into four main sections based on the research hypotheses previously mentioned. The first section concerned socio-demographics and included 11 components. The second section pertained to the residential environment and included 25 components for neighborhood facilities, 5 components for environmental features, 5 components for social environment, 8 components for housing conditions, 22 components for housing features, and 6 components for housing support services. The third section focused on local historical housing renovation rules, which had five components. The fourth section concerned sense of place, which was broken down into three sub-sections, place identity, place attachment, and place dependence, and had four items for each sub-section, as shown in Appendix A, Table A2.

Before conducting the technical modeling analysis, principal component analysis was used to derive all indexes. The internal reliability of all questions was also assessed using Cronbach's alpha. In general, the average measurement was found to be above 0.85, indicating the reliability of the scale. Unlike previous residential satisfaction studies that used multiple regression analysis [9], this study utilized logit models due to the presence of a binary dependent variable. Multiple linear regression (e.g., OLS technique) can produce predictive values outside a range of 0 and 1 in such cases, but logit models provide more robust results through the use of logistic regression methodology [39]. Additionally, unlike

earlier studies [8,9] that relied on the magnitude of coefficients to rank determinants, this study employed a bootstrap ranking LASSO logistic regression to rank determinants by importance for prediction and to reduce overfitting. According to Courville and Thompson [32], relying on the magnitude of coefficients is not reliable due to the presence of inter-correlation between determinants.

*4.3. Descriptive Analysis*

Table 2 illustrates the socio-demographic profile of respondents. In the study, 51.94% of the respondents were male and 48.06% were female. Most of the respondents were between the ages of 31 and 61 (53.49%), followed by those between 19 and 30 years old (27.91%), those over 61 (15.50%), and those under 18 (3.10%). The monthly income for 41.09% of respondents was between TRY 2000 and 3000, 23.26% earned between TRY 3000 and 4000, 19.38% earned less than TRY 2000, 10.08% earned between 4000 and 6000 lira, and 6.19% earned more than TRY 6000. Further, 51.16% of respondents were renters, while 48.84% were owners. The length of time respondents had lived in the housing area varied, with 21.71% living there for 10–20 years, 19.38% living there for 20–40 years, 13.95% living there for 5–10 years, and 20.16% living there for more than 40 years. The ethnic composition of the area was 65.12% Turkish-Cypriot and 34.88% non-Turkish-Cypriot. The household sizes of respondents varied, with 3.88% having one resident, 24.81% having two residents, 23.26% having three residents, 29.46% having four residents, and 18.59% having more than four residents. Most of the respondents were married (62.01%), while 37.99% were single. The highest education levels for the respondents were university degree (33.34%) and high school (33.33%), while some had only attended primary school (17.83%) or middle school (15.50%). The majority of respondents were employed in the public sector (87.60%), while 12.40% worked in the private sector. The most common number of bedrooms in respondents' houses was three (41.86%), while the least common was over four (1.55%). Overall, the descriptive analysis shows that the most satisfied respondents were female (54.22%), between 31 and 61 years old (55.42%), with monthly income between TRY 2000 and 3000 (43.37%), home ownership (57.83%), and length of residency between 10 and 20 years (24.10%). They were also Turkish-Cypriot (68.67%), with a household size of four residents (30.12%), married (63.85%), holding a university degree (36.14%), working in public employment (85.54%), and living in houses with three bedrooms (38.55%).

The descriptive analysis in Table 3 shows that residents' overall satisfaction with neighborhood facilities, environmental features, social environment, housing condition, local historical housing renovation rules, and sense of place was lower than the moderate level (3.14, 2.85, 3.08, 3.14, 2.91, and 3.42, respectively). Meanwhile, residents' satisfaction with overall housing features was at a moderate level (3.50) and satisfaction with overall housing support services was below the moderate level (3.03). This is consistent with previous studies [9,14,83]. The present study used a point of 3.5 as the moderate level of satisfaction, similar to previous studies [9,14].

**Table 3.** Overall satisfaction.

| Overall Satisfaction | Very Dissatisfied (%) | Dissatisfied (%) | Slightly Satisfied (%) | Satisfied (%) | Very Satisfied (%) | Mean | SD |
|---|---|---|---|---|---|---|---|
| Neighborhood facilities | 0.00 | 0.78 | 37.98 | 55.81 | 5.43 | 3.14 | 0.11 |
| Environmental features | 4.65 | 21.71 | 26.36 | 39.53 | 7.75 | 2.85 | 0.19 |
| Social environment | 4.65 | 12.40 | 25.58 | 46.51 | 10.85 | 3.08 | 0.19 |
| Housing conditions | 1.55 | 12.40 | 29.46 | 40.31 | 16.28 | 3.14 | 0.19 |
| Housing features | 0.00 | 3.10 | 20.16 | 60.47 | 16.28 | 3.50 | 0.14 |
| Housing support services | 3.10 | 9.30 | 37.98 | 44.19 | 5.43 | 3.03 | 0.16 |
| Local historical housing renovation rules | 3.88 | 8.53 | 49.61 | 32.56 | 5.43 | 2.91 | 0.16 |
| Sense of place | 0.00 | 0.78 | 25.78 | 62.50 | 10.94 | 3.42 | 0.11 |

*4.4. Econometrical Test Models*

This study uses the actual–aspiration gap conceptual approach to build the econometrical test model because it allows researchers to measure the gap between the actual experiences or needs of residents and their aspirations or expectations. By examining this gap, researchers can better understand how satisfied residents are with their living situation and identify any areas where improvements could be made. The use of this approach helps the study to gain a more comprehensive understanding of residential satisfaction and its determinants. The model includes four variables: socio-demographics, the residential environment index, the local historical housing renovation rules index, and the SOP index. The dependent variable, satisfaction, is measured by using a dummy variable that takes on a value of 1 if the respondent is satisfied and 0 if they are not. The other variables are measured using a 5-point Likert scale, with 1 representing the lowest level of satisfaction or agreement and 5 representing the highest. The model is represented by Equation (1), where $\alpha$ represents the coefficients for each variable and $\varepsilon$ represents the error term.

$$\begin{aligned} Satisfaction \quad = \quad & \alpha_0 + \alpha_1 Socio\ demographic + \alpha_2 Residential\ environment\ index \\ & + \alpha_3 Historical\ housing\ renovation\ rules\ index + \alpha_4 SOP\ index + \varepsilon \end{aligned} \tag{1}$$

In addition, the present study performs Equation (2) to test the effect of both sub-pillars of residential environment and SOP on RS. Therefore, this study replaces the residential environment index with the sub-indices of neighborhood facilities, environmental features, social environment, housing conditions, housing features, and housing support services. It also replaced the SOP index with PI, PA, and PD sub-indices.

$$\begin{aligned} Satisfaction \quad = \quad & \alpha_0 + \alpha_1 Socio\ demographic + \alpha_2 \sum Residential\ environment\ sub\ index \\ & + \alpha_3 Historical\ housing\ renovation\ rules\ index + \alpha_4 \sum SOP\ sub\ index + \varepsilon \end{aligned} \tag{2}$$

This study also performs Equation (3) to examine the effect of components of each pillar on RS.

$$\begin{aligned} Satisfaction \quad = \quad & \alpha_0 + \alpha_1 Socio\ demographic + \alpha_2 \sum Residential\ environment\ components \\ & + \alpha_3 \sum Historical\ housing\ renovation\ rules\ components + \alpha_4 \sum SOP\ components + \varepsilon \end{aligned} \tag{3}$$

Similar to Equation (1), the present study also uses a dummy variable for measuring satisfaction in both Equations (2) and (3). In addition, it measures the determinants using the above-mentioned 5-point Likert scales.

## 5. Empirical Results

Before performing the technical modeling analysis, all indexes were derived using principal component analysis (PCA). Cronbach's alpha was examined for the internal reliability of all questions. In general, we found that the average of this measurement is above 0.85, which shows the scale reliability.

*Estimation Results*

The results of the logit regression analysis, presented in Table 4, indicate that the residential environment, local historical housing renovation rules, and sense of place (SOP) pillars have a statistically significant positive impact on residents' satisfaction (RS). The marginal effects of the logit regression show that an increase of one unit in the residential environment, local historical housing renovation rules, and SOP pillars leads to a 13%, 9%, and 36% increase, respectively, in the likelihood of residents' satisfaction. Model 2 of the analysis supports previous studies [9,65] in finding that the sub-pillars of the residential environment, specifically environmental features, social environment, and housing support services, significantly determine RS. Additionally, the results confirm earlier research [108,109] in showing that the sub-pillars of SOP, particularly place identity (PI) and place attachment (PA), have a significant impact on RS. These findings are not surprising given that the historic urban quarter of the Walled City has been preserved by

cultural heritage programs. Studies have shown that the protection of cultural heritage can enhance residents' sense of place ([100,102].

**Table 4.** Pillars and sub-pillars of residential satisfaction.

| Variables | Model 1 | | Model 2 | |
|---|---|---|---|---|
| | Coefficient Estimates | Marginal Effects | Coefficient Estimates | Marginal Effects |
| Socio-demographic characteristics | | | | |
| Gender | 0.69 | 0.13 | 1.62 | 0.11 |
| | (1.24) | (1.29) | (1.17) | (1.25) |
| Age | −0.40 | −0.08 | −1.36 | −0.09 |
| | (−0.76) | (−0.74) | (−1.18) | (−1.28) |
| Monthly income | −0.12 | −0.02 | −0.10 | −0.006 |
| | (−0.38) | (−0.38) | (−0.08) | (−0.07) |
| Home ownership | −0.47 | −0.09 | 1.40 | 0.09 |
| | (−0.72) | (−0.72) | (1.49) | (0.78) |
| Length of residency | −0.24 | −0.04 | −0.81 ** | −0.05 *** |
| | (−1.14) | (−1.17) | (−2.44) | (−1.75) |
| Ethnicity | −0.34 | −0.06 | −2.39 ** | −0.15 * |
| | (−0.70) | (−0.70) | (−2.14) | (−2.67) |
| Household size | −0.30 | −0.06 | −1.55 *** | −0.10 * |
| | (−0.98) | (−1.02) | (−1.95) | (−2.31) |
| Marital status | 0.20 | 0.04 | 2.34 *** | 0.15 * |
| | (0.26) | (0.26) | (1.65) | (2.23) |
| Education | 0.21 | 0.03 | 0.31 | 0.02 |
| | (0.81) | (0.78) | (0.63) | (0.57) |
| Employment sector | −0.50 | −0.08 | −1.34 | −0.06 |
| | (−0.52) | (−0.57) | (−0.72) | (−0.52) |
| Number of bedrooms | 0.51 | 0.10 | 0.59 | 0.03 |
| | (1.27) | (1.36) | (0.71) | (0.65) |
| Residential environment | 0.67 ** | 0.13 ** | — | — |
| | (2.47) | (2.35) | | |
| Neighborhood facilities | — | | 1.14 | 0.07 |
| | | | (1.17) | (1.15) |
| Environmental features | — | — | 1.10 * | 0.07 *** |
| | | | (2.19) | (1.69) |
| Social environment | — | — | 1.19 *** | 0.08 *** |
| | | | (1.67) | (1.87) |
| Housing conditions | —- | — | −0.81 | −0.05 |
| | | | (−0.92) | (−1.13) |
| Housing features | —- | — | 0.70 | 0.04 |
| | | | (1.05) | (1.54) |
| Housing support services | —- | — | 3.76 *** | 0.25 * |
| | | | (1.75) | (2.12) |
| Local historical housing renovation rules | 0.47 *** | 0.09 *** | 2.10 * | 0.14 * |
| | (1.65) | (1.69) | (2.71) | (2.01) |
| Sense of place | 1.86 * | 0.36 * | — | — |
| | (4.88) | (5.45) | | |
| Place identity | — | — | 2.47 ** | 0.16 * |
| | | | (2.21) | (3.13) |
| Place attachment | — | — | 2.47 *** | 0.15 * |
| | | | (1.65) | (2.15) |
| Place dependence | — | — | 0.27 | 0.01 |
| | | | (0.26) | (0.20) |
| Pseudo R$^2$ | 0.44 * | — | 0.76 * | — |

Note: All the pillars and sub-pillars are indexed-based. In this table: * Significant at 0.01 level. ** Significant at 0.05 level. *** Significant at 0.10 level.

Examining the socio-demographic characteristics of the respondents, the results suggest that length of residency and household size have a negative effect on predicting RS. These findings are consistent with recent studies [23,25] that found negative and significant effects for these variables. The results also indicate that ethnicity and marital status have a negative and positive impact on RS, respectively. This suggests that Turkish-Cypriot and married residents are more satisfied compared to non-Turkish and single residents. The results also reveal that the predictor variables explain 76% of the variance in resident satisfaction (RS). As emphasized by Fang [54], it is important for local authorities, housing policy decision makers, and urban planners to identify the essential elements of RS in order to improve residential environments that meet the current needs of residents.

According to Figure 6, the order of importance of variables on RS in this study was determined using a bootstrap LASSO logistic regression model (the variable importance order results are extracted after 10,000 times bootstrapping with the binomial kernel function. In the LASSO estimation, the optimal Lambda (0.068) is defined using k-fold cross-validation. We set number of folds to 8 with deviance as a measure of loss to use for cross-validation.). The results show that housing support services are the most important factor for RS, followed by PA, PI, gender, employment sector, and so on. The remaining variables, listed in order of importance, are social environment, PD, local historical housing renovation rules, household size, home ownership, marital status, environmental features, number of bedrooms, age, housing conditions, ethnicity, housing features, length of residency, neighborhood facilities, education, and income. These findings suggest that among the various factors studied, housing support services have the greatest influence on residents' satisfaction.

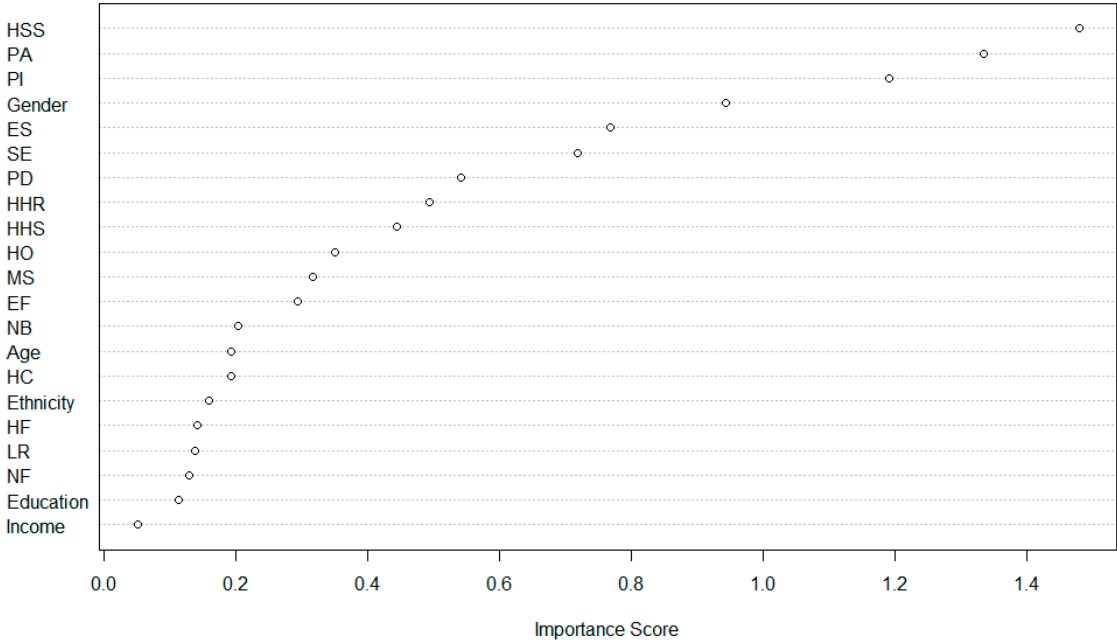

**Figure 6.** Variable selection using Bootstrap ranking LASSO logistic regression model. Note: Variable definitions: Housing Support Services (HSS), Place Attachment (PA), Place Identity (PI), Employment Sector (ES), Social Environment (SE), Place Dependence (PD), Local Historical Housing Renovation Rules (HHR), Household Size (HHS), Home Ownership (HO), Marital Status (MS), Environmental Features (EF), Number of Bedrooms (NB), Housing Conditions (HC), Housing Features (HF), Length of Residency (LR), Neighborhood Facilities (NF).

Table 5 shows the logit regression results of Equation (3). However, for the sake of brevity, only the significant results are presented here. The results suggest that 27 of the components significantly contributed to predicting RS. Among the components, eight

components with negative signs and nineteen components with positive signs impact RS. Overall, nineteen of these significant components are linked to the residential environment, followed by one component for local historical housing renovation rules, and seven components for SOP pillars. In addition, Table 5 shows that the residential environment sub-pillars (three components of neighborhood facilities; eight components of environmental features, social environment, housing conditions, and housing support services; and eight components of housing features) significantly impact RS. The findings support a study by Vehbi and Hoskara [28], which showed that the current needs of the residents are not fulfilled by physical conditions in the historic urban quarters.

**Table 5.** Components of residential satisfaction.

| Components | Model 3 | | | |
| --- | --- | --- | --- | --- |
| | Coefficient Estimates | *t*-Value | Marginal Effects | *t*-Value |
| Shopping center | −0.66 *** | (−1.94) | −0.14 * | (−2.09) |
| Namik Kemal Square | 1.09 * | (3.48) | 0.23 * | (3.40) |
| Public parking | 0.93 * | (2.74) | 0.19 * | (2.93) |
| Traffic density | −0.45 ** | (−2.36) | −0.10 ** | (−2.40) |
| Night lighting | −0.34 *** | (−1.67) | −0.07 *** | (−1.67) |
| Neighborhood relations | 0.30 *** | (1.75) | 0.07 *** | (1.67) |
| Security | 0.27 ** | (2.15) | 0.06 *** | (1.65) |
| Quality of windows | 0.64 ** | (2.04) | 0.13 ** | (2.03) |
| Lighting of stairwell | 0.46 ** | (2.02) | 0.09 ** | (2.04) |
| Location of kitchen | 1.14 * | (2.83) | 0.24 ** | (2.57) |
| Location of toilet | −1.07 ** | (−2.10) | −0.23 ** | (−2.03) |
| Size of living room | −0.70 *** | (−1.86) | −0.15 *** | (−1.79) |
| Size of kitchen | 0.96 ** | (2.55) | 0.20 ** | (2.50) |
| Natural light of living room | 0.59 *** | (1.68) | 0.13 *** | (1.75) |
| Natural light of kitchen | −1.20 ** | (−2.26) | −0.25 ** | (−2.25) |
| Natural ventilation of bedroom | 1.25 ** | (2.43) | 0.27 ** | (2.21) |
| Thermal isolation | −0.77 * | (−3.09) | −0.16 * | (−3.13) |
| Electrical repairs | 0.64 ** | (2.07) | 0.14 ** | (2.07) |
| Internet | 0.79 * | (2.82) | 0.17 * | (2.77) |
| Height of building | 0.31 *** | (1.96) | 0.07 *** | (1.65) |
| Identity 1 | 1.04 * | (3.62) | 0.211 * | (4.15) |
| Identity 4 | 1.13 * | (3.78) | 0.23 * | (3.82) |
| Attachment 1 | 1.11 * | (2.77) | 0.22 * | (2.80) |
| Attachment 3 | 1.16 * | (2.89) | 0.23 * | (2.80) |
| Dependence 1 | 0.50 ** | (2.25) | 0.09 ** | (2.33) |
| Dependence 2 | 0.47 ** | (2.11) | 0.09 ** | (2.11) |
| Dependence 4 | −0.94 * | (−3.86) | −0.18 * | (−2.17) |

Note: This table presents only the significant components of residential satisfaction. * Significant at 0.01 level. ** Significant at 0.05 level. *** Significant at 0.10 level.

## 6. Discussion

To the best of the authors' knowledge, this study is among the first to specifically examine the predictors of RS in the Walled City of Famagusta. While Davoodi and Dağlı [29] identified important factors related to the sustainability of the Walled City through a "Marginal improvement priority" approach, they did not measure the gap between actual reality and aspirations through resident satisfaction levels, which this study tried to achieve. To create an empirical satisfaction model for the historical area of the Walled City, this study applies the actual–aspiration gap theory, as proposed by Galster [5], which allows researchers to measure the gap between the actual experiences or needs of residents and their aspirations or expectations.

The descriptive findings show that, although residents are just over slightly satisfied with the neighborhood facilities, social environment, housing conditions, housing support services, and SOP, they are slightly dissatisfied with the environmental features and local historical housing renovation rules. Further, findings show that residents are moderately

satisfied with housing features. Focusing on the empirical findings, the logit estimation results reveal that the residential environment, local historical housing renovation rules, and SOP have statistically significant and positive impacts on RS. Results suggest that improvements in RS in the historic urban quarters depend not only on the residential environment, as historical housing renovation rules and SOP are also significant pillars of RS. More specifically, empirical results show that the betterment of environmental features, social environment, housing support services, PI, and PA can significantly enhance resident satisfaction. The findings also indicate that length of residency, household size, and ethnicity with negative signs and marital status with a positive sign significantly impact RS. Moreover, the results of the bootstrap LASSO logistic regression model suggest that housing support services are the essential factor of RS. This is followed by PA, PI, gender, and employment sector.

Additionally, there are some shortcomings of the research because of the methodology chosen to collect the data. As mentioned earlier in the Data Collection and Methodology section, the conveying sampling method used minimizes the generalization of the findings to other locations. Furthermore, using a very unique location as a case study reduces the external validity of the study, which adds to the problem of generalization. Consequently, if similar research is conducted in areas with similar attitudes, such as the Walled City of Nicosia and the old quarters of cities, the external validity of the results could have been increased.

## 7. Conclusions

Although several studies have empirically investigated the predictors of RS in public and private housing, there is less attention in the empirical literature on investigating the determinants of RS, specifically in the historic environment of cities in developing countries. Therefore, this study, by considering the specific and distinguishing characteristics of historical areas, attempted to provide an empirical satisfaction model for exploring the determinants of RS in a historical area of the Walled City, Famagusta. This study has a distinctive contribution by the inclusion of different determinants of RS and employing a bootstrap ranking LASSO logistic regression for ranking the importance of determinants.

This study also provides valuable insight into the needs and expectations of residents in historic urban quarters and may be useful to housing policy decision makers, urban planners, and municipalities in improving the residential environment to better meet these needs in historic areas. The results of this study also contribute to the literature on residential satisfaction and provide a foundation for further research on applying the empirical satisfaction model used in this study to other historical areas in order to identify determinants of residential satisfaction. Furthermore, we believe that RS in historical areas can be important for sustainability. The results can be considered as a crucial step in the sustainable development process, and identifying the significant determinants of RS helps to completely fulfill the contemporary needs of residents in other physically degraded and urban-fabric-deteriorated historical environments. High levels of residential satisfaction can help to foster a sense of community and social cohesion, which can, in turn, support efforts to promote sustainable behaviors and practices within the community. Furthermore improving residential satisfaction in historical areas can also help to support the preservation and revitalization of these areas, which can be important for maintaining the cultural and historical character of a city or region.

Although this paper provides empirical findings for modeling the predictors of RS in historic urban quarters, further studies should be conducted in more historical environments to achieve generalization of the findings of this research to other locations.

**Author Contributions:** Conceptualization, T.D., B.Y. and U.U.D.; methodology, T.D.; software, T.D.; validation, T.D., B.Y. and U.U.D.; formal analysis, T.D.; investigation, T.D.; resources, T.D.; data curation, B.Y.; writing—original draft preparation, T.D.; writing—review and editing, B.Y.; visualization, T.D. and B.Y. All authors have read and agreed to the published version of the manuscript.

**Funding:** This research received no external funding.

**Institutional Review Board Statement:** Not applicable.

**Informed Consent Statement:** Not applicable.

**Data Availability Statement:** The data that support the findings of this study are available from the corresponding author upon reasonable request.

**Conflicts of Interest:** The authors declare no conflict of interest.

## Appendix A. Summary of Existing Literature on RS and Questionnaire

**Table A1.** Summary of residential satisfaction studies.

| Authors | Case Studies | Selected Factors | Findings |
|---|---|---|---|
| [69] | Households sampled (Wooster, Ohio) | Context of dwelling and neighborhood, characteristics of households. | Characteristics of households and contextual characteristics of the dwellings and neighborhood have significant effect on residential satisfaction. |
| [64] | Public housing units (Abuja, Nigeria) | Structure types, building features, housing condition, neighborhood facilities, public housing management. | Residents are dissatisfied with all the factors except for the neighborhood facilities. |
| [17] | Modern and old neighborhood (Edirne, Turkey) | Perceived living conditions, physical surrounding, social relations, local authorities, quality of the facilities. | Perceived attitude toward residential satisfaction with social relations, and the local authorities are relatively higher in a modern neighborhood. |
| [65] | Mass housing areas (Istanbul, Turkey) | Accessibility, inhabited residential environment, various facilities in the inhabited environment, environmental security, neighbor relationships, appearance of housing environment. | Accessibility to open areas is statistically significant with positive signs and has the highest impact on housing and environmental quality satisfaction. |
| [20] | Private low-cost housing (Terengganu, Malaysia) | Dwelling unit features, housing services, neighborhood facilities and environment. | Degrees of residential satisfaction are higher with dwelling units and services provided by the developers than neighborhood facilities and environment. |
| [22] | Neighborhoods (Prenestino-Labicano, Torre Angela, Rome) | Physical attributes of the environment, cognitive perceptions and affective appraisals of residents, urban activities, socio-demographic. | Cognitive, affective, and urban activities are significant predictors of residential satisfaction. |
| [44] | Newly designed public low-cost housing (Kuala Lumpur, Malaysia) | Dwelling unit features, dwelling unit support services, public facilities, social environment, neighborhood facilities. | Residents moderately satisfied with neighborhood facilities, support services, and public facilities more than dwelling unit features and social environment. |
| [18] | Planned community (Elenbrook, Western Australia) | Sense of community, sense of belonging, and sense of place. | Sense of community, sense of belonging, and sense of place have a positive nexus with residential satisfaction. |

**Table A1.** *Cont.*

| Authors | Case Studies | Selected Factors | Findings |
|---|---|---|---|
| [40] | Public housing (Ogun, Nigeria) | Dwelling unit features, dwelling unit support services, neighborhood environment, management of housing estates, housing acquisition process, social environment. | Level of residential satisfaction is comparatively higher with dwelling unit features than neighborhood facilities and services and respondents are dissatisfied with the housing conditions. |
| [8] | Public core housing (Abeokuta, Ogun State, Nigeria) | Socio-economic characteristics, housing unit characteristics, neighborhood facilities and environment, management and services. | Socio-economic characteristics, housing unit characteristics, neighborhood facilities and environment, management and services are significant predictors of residential satisfaction. |
| [66] | Squatter houses, and apartment buildings (Dikmen, Ankara) | Architectural features, interior and economic features of the house, functionality and location of the house, and social features of the housing environment. | Housing satisfaction is a multidimensional phenomenon containing physical, social and economic dimensions. |
| [15] | Public housing (Hangzhou, China) | Housing characteristic, neighborhood characteristic, public facilities, social environment, residence comparison, housing allocation scheme, residents' characteristics. | Neighborhood characteristic, public facilities, housing characteristics, public housing allocation scheme, socio-economic, and residence comparison factors impact on residential satisfaction. |
| [9] | Double-storey terrace housing (Kuala Lumpur, Malaysia) | Public facilities, social environment, neighborhood facilities, housing support services, physical features. | Improvements of housing design and neighborhood elements can significantly enhance the residents' overall housing satisfaction. |
| [25] | Public housing (Shiraz, Iran) | Physical features, public facilities, public services, social participation and cohesion. | Physical features variable is the main determinants of overall residential satisfaction. |
| [19] | Public housing (Tehran, Iran) | Place identity, place dependence, place attachment (sense of place components). | Sense of place components are positive predictive factors for residential satisfaction. |

Note: This table shows the summary of previous residential satisfaction studies.

**Table A2.** The sample questionnaires of sense of place.

| A: Place identity | | | | |
|---|---|---|---|---|
| Everything about my residential area (Walled city) is a reflection of me. | | | | |
| Strongly disagree | Disagree | Do not know | Agree | Strongly agree |
| My residential area (walled city) says very little about who I am. | | | | |
| Strongly disagree | Disagree | Do not know | Agree | Strongly agree |
| I feel that I can really be myself at my residential area (Walled city). | | | | |
| Strongly disagree | Disagree | Do not know | Agree | Strongly agree |
| My residential area (Walled city) reflects the type of person I am. | | | | |
| Strongly disagree | Disagree | Do not know | Agree | Strongly agree |
| **B: Place attachment** | | | | |
| I feel relaxed when I'm at my residential area (Walled city). | | | | |
| Strongly disagree | Disagree | Do not know | Agree | Strongly agree |
| I feel happiest when I'm at my residential area (Walled city). | | | | |
| Strongly disagree | Disagree | Do not know | Agree | Strongly agree |
| My residential area (Walled city) is my favorite place to be | | | | |
| Strongly disagree | Disagree | Do not know | Agree | Strongly agree |
| I really miss my residential area (Walled city) when I'm away from it for too long. | | | | |
| Strongly disagree | Disagree | Do not know | Agree | Strongly agree |
| **C: Place dependence** | | | | |

**Table A2.** *Cont.*

| A: Place identity | | | | |
|---|---|---|---|---|
| My residential area (Walled city) is the best place for doing the things that I enjoy most. | | | | |
| Strongly disagree | Disagree | Do not know | Agree | Strongly agree |
| For doing the things that I enjoy most, no other place can compare to my residential area (Walled city). | | | | |
| Strongly disagree | Disagree | Do not know | Agree | Strongly agree |
| My residential area (Walled city) is not a good place to do the things I most like to do. | | | | |
| Strongly disagree | Disagree | Do not know | Agree | Strongly agree |
| As far as I am concerned, there are better places to be than at my residential area (Walled city). | | | | |
| Strongly disagree | Disagree | Do not know | Agree | Strongly agree |

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
