# Peer review of "Measuring Residential Satisfaction in Historic Areas Using Actual–Aspiration Gap Theory: The Case of Famagusta, Northern Cyprus"

_sustainability, doi:10.3390/su15053917_

Round 1
Reviewer 1 Report
I am familiar with the literature on RS, and I agree with the authors' assertion that there is little research on this front in the realm of historic properties.
I also believe that the issue of RS is an appropriate part of a sustainable community, and thus is a good fit for this journal.
For the SOP pillars, can you please explain the difference between such things as "place attachment", "place dependence", and "place identity"? It is not clear to me how these differ. It looks like you try and explain further on this front on page 8, but possibly give further examples to the reader on how these three concepts differ from one another. A tangible example here might go a long way for the reader.
Because much of her introduction incorporated previous literature on RS, it seems like it might have also been OK to include it in the lit review section.
I felt that the authors provided a pretty thorough job on the lit review, so no problem there. In areas where there were inconsistent effects of predictors of RS, this is a good indication that certain attributes have differing effects on RS across cultures. I also appreciated the authors providing an appendix on all of the research out there on RS.
You mention that you do not know of any research on RS that has occurred in the Walled City of Famagusta, North Cyprus, but can't you also say that it is the first (or among the first) RS study on historical housing in general?
In your section on socio-demographic characteristics and their effects on RS, you state that age, sex, etc. have effects on RS in certain studies...can you elaborate further on how?
One thing that I don't quite understand about H3 is how does one find differentiation among historic preservation and designation rules. Aren't the rules the same across all areas of the historic site? If so, you would not find any differentiation across space, and your research on this front would be somewhat worthless on this aspect. Please explain further.
Although I am pretty well-schooled, I believe that this is the first time I have ever heard the word "conative". Perhaps fleshing out this definition would be helpful as well.
In Figure 2, I am assuming that "housing renovation rules" is the same thing as "historical housing/buildings", correct?
I also appreciated the authors' use of LASSO to prevent overfitting and and ranking determinants by importance.
It looks like the data collection and methodologies section (4.2) was repeated....this was probably just an oversight.
Not that it is the biggest thing, but the descriptive statistics described in Tables 2 and 3 are not exactly "empirical results"...it should probably go right before that section. This was something that was drilled into my head earlier in my career. Feel free to push back on this, though, if you feel I am wrong on this.
Also regarding these tables, do you have any info on the general population here, just to make sure that you have a representative sample that does not bias your results?
Author Response
Reviewer 1: Comments and Suggestions for Authors
1. I am familiar with the literature on RS, and I agree with the authors' assertion that there is little research on this front in the realm of historic properties.
Authors: Thank you for your valuable comment.
2. I also believe that the issue of RS is an appropriate part of a sustainable community, and thus is a good fit for this journal.
Authors: Thank you for your valuable comment.
3. For the SOP pillars, can you please explain the difference between such things as "place attachment", "place dependence", and "place identity"? It is not clear to me how these differ. It looks like you try and explain further on this front on page 8, but possibly give further examples to the reader on how these three concepts differ from one another. A tangible example here might go a long way for the reader.
Authors: Thank you for your valuable comments. Authors, tried to make more clear the difference between "place attachment", "place dependence", and "place identity". In this current, Authors explained that intangible elements refer to the socio-cultural factors (i.e. activities that occur in the historical places with regards to the traditions) which this study emphasized. Please see Section 3.1.4.1.
4. Because much of her introduction incorporated previous literature on RS, it seems like it might have also been OK to include it in the lit review section.
Authors: Thank you. As mentioned in our previous respond we decided that it is more appropriate to put the definitions under SOP section 3.1.4.1.
5. I felt that the authors provided a pretty thorough job on the lit review, so no problem there. In areas where there were inconsistent effects of predictors of RS, this is a good indication that certain attributes have differing effects on RS across cultures. I also appreciated the authors providing an appendix on all of the research out there on RS.
Authors: Thank you for your valuable comments
6. You mention that you do not know of any research on RS that has occurred in the Walled City of Famagusta, North Cyprus, but can't you also say that it is the first (or among the first) RS study on historical housing in general?
Authors: Thanks for this valuable suggestion. Therefore, Authors mentioned, “This study is among the first to specifically examine the predictors of RS in the Walled City of Famagusta”. Please see Section 1 with line number of (78- 79).
7. In your section on socio-demographic characteristics and their effects on RS, you state that age, sex, etc. have effects on RS in certain studies...can you elaborate further on how?
Authors: This is a very good observation. In line with this comment, Authors explained the reason of selected determinants of socio-demographic characteristics such as (e.g., age, martial). Please see Section 3.1.1
8. One thing that I don't quite understand about H3 is how does one find differentiation among historic preservation and designation rules. Aren't the rules the same across all areas of the historic site? If so, you would not find any differentiation across space, and your research on this front would be somewhat worthless on this aspect. Please explain further.
Authors: Thank you for your valuable comment. The rules of Walled City, Famagusta is different than across the area. Therefore, Authors, added to examine the effect of local historical housing renovation rules on RS, this study collected information on existing housing renovation rules of the “Walled City” from the department of Antiquities law in the Turkish Republic of North Cyprus (TRNC). Please see Section 3.1.3
9. Although I am pretty well-schooled, I believe that this is the first time I have ever heard the word "conative". Perhaps fleshing out this definition would be helpful as well.
Authors: Thank you for this observation. However, literature on RS and PD discusses conative link. Therefore, rather than contradicting the existing literature, the authors in Section 3.1.4.1 added behavioral links as a synonym of conative link.
10. In Figure 2, I am assuming that "housing renovation rules" is the same thing as "historical housing/buildings", correct?
Authors: Thanks for this observation. Authors, changed “Housing renovation rules” to “Historical housing renovation rules”. Please see Section 3.2. Figure 2.
11. I also appreciated the authors' use of LASSO to prevent overfitting and ranking determinants by importance.
Authors: Thank you for this observation.
12. It looks like the data collection and methodologies section (4.2) was repeated.... this was probably just an oversight.
Authors: Thank you for your valuable observation. Authors, corrected this mistake and deleted the repeated section.
13. Not that it is the biggest thing, but the descriptive statistics described in Tables 2 and 3 are not exactly "empirical results"...it should probably go right before that section. This was something that was drilled into my head earlier in my career. Feel free to push back on this, though, if you feel I am wrong on this.
Authors: Thank you for your valuable comment. Therefore, Authors combined Section 5.1 with Section 5.2 and changed to the Section 4.3 number as “Descriptive Analysis”. Also, Section 4.3 number changed to Section 4.4. Please see Section 4.3
14. Also regarding these tables, do you have any info on the general population here, just to make sure that you have a representative sample that does not bias your results?
Authors: Thanks for this observation. Authors added the total number of the historical houses which can be compared with the total number of houses (one member for one household) Please see Section 4.2
Reviewer 2 Report
The study examines the factors affecting residential satisfaction (RS) in historic areas.
The scientific background is clearly outlined. The literature review is well structured. The citations are correct and well attributed to the various trend topics identified. The methodology is clear and precisely described. The results are well presented.
The study investigates four pillars (Socio-demographic characteristics; Residential environment; Local historical housing renovation rules; Sense of place) which have a humanistic nature and are usually studied by sociologists, anthropologists, urban planners, etc. The research associates these pillars with mathematical models that allow the description of their impact dynamics on RS. In this sense the final results appear acceptable from a scientific point of view.
In conclusion I would like to suggest:
1) To consider in the study of the Sense of Place (SOP) what Christian Norberg-Schulz argues in the volume "Genius Loci: Towards a Phenomenology of Architecture" (1980) Academy Editions Ltd (a division of John Wiley & Sons Ltd.)
2) To specify the meaning of what is written in lines 628, 629, 630. As written, it does not seem convincing and should be supported by some scientific data analysis.
Author Response
The study examines the factors affecting residential satisfaction (RS) in historic areas.
The scientific background is clearly outlined. The literature review is well structured. The citations are correct and well attributed to the various trend topics identified. The methodology is clear and precisely described. The results are well presented.
Authors: Thank you for this observation.
The study investigates four pillars (Socio-demographic characteristics; Residential environment; Local historical housing renovation rules; Sense of place) which have a humanistic nature and are usually studied by sociologists, anthropologists, urban planners, etc. The research associates these pillars with mathematical models that allow the description of their impact dynamics on RS. In this sense the final results appear acceptable from a scientific point of view.
Authors: Thank you for this observation.
In conclusion I would like to suggest:
1) To consider in the study of the Sense of Place (SOP) what Christian Norberg-Schulz argues in the volume "Genius Loci: Towards a Phenomenology of Architecture" (1980) Academy Editions Ltd (a division of John Wiley & Sons Ltd.)
Authors: Thanks for your valuable comment. Therefore, Authors referred to this valuable book. Please see Section 3.1.4.1
2) To specify the meaning of what is written in lines 628, 629, 630. As written, it does not seem convincing and should be supported by some scientific data analysis.
Authors: Thanks for your valuable comment. Authors agree with your comment and deleted those lines. Moreover, authors revised the conclusion accordingly. Please see Section 7.
Reviewer 3 Report
This is a very interesting study, the methodology used is appropriate, and the data are primary data. However, there are still some areas that need further improvement and refinement.
1.The reference citation is not very standardized. At least the author needs to be mentioned at the citation in the manuscript, not in the following way:
For example, [3] defined RS as 32 the perception of the gap between residential expectation and reality, while [4] argued 33 that it depends on the compatibility between real housing satisfaction and housing norms.
Therefore, the authors are advised to adjust all such irregular literature citations.
2.In the hypothesis section, the authors propose the hypothesis about "There is a significant relationship", however, it is not clear whether it is a positive or negative correlation. It is suggested that in the hypothesis, the significant positive or significant negative correlation is presented directly.
3.In the sample description section, there is no information on what sampling method was used, the representativeness of the sample, etc. The authors need to add this section.
4.Since the first-hand data was obtained through a questionnaire, it is necessary to describe how the survey ensured data quality.
5.Before 6. conclusion, 1 section should be added for discussion, mainly to introduce the theoretical implications, practical implications, and research shortcomings of the study.
6.A major limitation of this study is that Walled City, Famagusta was used as the study population, but this is a unique location, so the study faces the threat of external validity, i.e., how the results of this study can be generalized to other locations and whether they are generalizable. This should be discussed and clarified by the authors.
Author Response
This is a very interesting study, the methodology used is appropriate, and the data are primary data. However, there are still some areas that need further improvement and refinement.
Authors: Thank you for this observation.
1.The reference citation is not very standardized. At least the author needs to be mentioned at the citation in the manuscript, not in the following way:
For example, [3] defined RS as 32 the perception of the gap between residential expectation and reality, while [4] argued 33 that it depends on the compatibility between real housing satisfaction and housing norms. Therefore, the authors are advised to adjust all such irregular literature citations.
Authors: Thank you for your valuable comments. Accordingly, authors corrected the irregularities
2.In the hypothesis section, the authors propose the hypothesis about "There is a significant relationship", however, it is not clear whether it is a positive or negative correlation. It is suggested that in the hypothesis, the significant positive or significant negative correlation is presented directly.
Authors: Thanks for your valuable comment. Accordingly, authors changed “There is a significant relationship" to “There is a positive or negative significant relationship between”. Please see Section 3.1.1, 3.1.2, 3.1.3, 3.1.4.2.
3.In the sample description section, there is no information on what sampling method was used, the representativeness of the sample, etc. The authors need to add this section.
Authors: Thanks for your valuable comment. Authors explained the sampling method, which is “convenience sampling technique”. Please see Section 4.2 with.
4.Since the first-hand data was obtained through a questionnaire, it is necessary to describe how the survey ensured data quality.
Authors: Thanks for your valuable comment. The authors at the beginning of the section 5 explained how we the survey ensured data quality. “Before performing the technical modeling analysis, all indexes were derived using principal component analysis (PCA). Cronbach’s alpha was examined for the internal reliability of all questions. In general, we found that the average of this measurement is above 0.85 which shows the scale reliability”. Please see Section 5.
5.Before 6. conclusion, 1 section should be added for discussion, mainly to introduce the theoretical implications, practical implications, and research shortcomings of the study.
Authors: Thanks for your valuable comment. Authors, added section 6 for the discussion before conclusion. Accordingly, Authors revised the conclusion as well. Please see Section 6 and Section 7.
6.A major limitation of this study is that Walled City, Famagusta was used as the study population, but this is a unique location, so the study faces the threat of external validity, i.e., how the results of this study can be generalized to other locations and whether they are generalizable. This should be discussed and clarified by the authors.
Authors: Thanks for your valuable comment. Authors, tried to clarify this in the discussion section. Please see Section 6 the last paragraph.
Round 2
Reviewer 3 Report
The paper meets the criteria for publication.